# Cell Plasticity of Marine Mediterranean Diazotrophs to Climate Change Factors and Nutrient Regimes

Víctor Fernández-Juárez [1,*], Elisa H. Zech [2], Elisabet Pol-Pol [3] and Nona S. R. Agawin [3]

1   Department of Biology, Marine Biological Section, University of Copenhagen, 3000 Helsingør, Denmark
2   Archaea Physiology & Biotechnology Group, Department of Functional and Evolutionary Ecology, Universität Wien, 1030 Wien, Austria
3   Marine Ecology and Systematics (MarES) Department of Biology, University of the Balearic Islands, 07122 Palma, Spain
*   Correspondence: victor.fj@bio.ku.dk

**Abstract:** Ocean acidification and warming are current global challenges that marine diazotrophs must cope with. Little is known about the effects of pH and temperature changes at elevated $CO_2$ levels in combination with different nutrient regimes on $N_2$ fixers, especially on heterotrophic bacteria. Here, we selected four culturable diazotrophs, i.e., cyanobacteria and heterotrophic bacteria, found in association with the endemic Mediterranean seagrass *Posidonia oceanica*. We tested different pH (from pH 4 to 8) and temperature levels (from 12 to 30 °C), under different nutrient concentrations of both phosphorus, P (0.1 μM and 1.5 mM), and iron, Fe (2 nM and 1 μM). We also tested different $CO_2$ concentrations (410 and 1000 particles per million (ppm)) under different P/Fe and temperature values (12, 18, and 24 °C). Heterotrophic bacteria were more sensitive to changes in pH, temperature, and $CO_2$ than the cyanobacterial species. Cyanobacteria were resistant to very low pH levels, while cold temperatures stimulated the growth in heterotrophic bacteria but only under nutrient-limited conditions. High $CO_2$ levels (1000 ppm) reduced heterotrophic growth only when cultures were nutrient-limited, regardless of temperature. In contrast, cyanobacteria were insensitive to elevated $CO_2$ levels, independently of the nutrient and temperature levels. Changes in $N_2$ fixation were mainly controlled by changes in growth. In addition, we suggest that alkaline phosphatase activity (APA) and reactive oxidative species (ROS) can be used as biomarkers to assess the plasticity of these communities to climate change factors. Unlike other studies, the novelty of this work lies in the fact that we compared the responses of cyanobacteria vs. heterotrophic bacteria, studying which changes occur at the cell plasticity level. Our results suggest that the responses of diazotrophs to climate change may depend on their P and Fe status and lifestyle, i.e., cyanobacteria or heterotrophic bacteria.

**Keywords:** phenotypic plasticity; ocean acidification and warming; *Posidonia oceanica*; $CO_2$; nutrients; diazotrophs; alkaline phosphatase activity (APA); reactive oxygen species (ROS)





## 1. Introduction

Carbon dioxide ($CO_2$) emissions have been increasing since the start of the industrial revolution in 1750 [1]. Anthropogenic activities will continuously emit $CO_2$ into the atmosphere, and $CO_2$ concentration is expected to reach between 750 and 1000 particles per million (ppm) by the end of this century from the current atmospheric levels of 418 ppm (National Oceanic and Atmospheric Administration (NOAA), https://www.esrl.noaa.gov/gmd/ccgg/trends/weekly.html, accessed on 15 January 2023) [2]. Increasing $CO_2$ levels have clear consequences on the chemistry of marine environments, as one-third of the anthropogenic emissions are accumulated in the oceans [3]. Therefore, the accumulation of $CO_2$, which leads to the formation of carbonic acid ($H_2CO_3$) and then bicarbonate ($HCO_3^-$), releasing protons ($H^+$), causes ocean acidification. By the end of this century, it is predicted that pH will reach values of 7.7 from current pH values of 8–8.25 [4]. Increasing $CO_2$

emissions also cause changes in the Earth's energy heat balance, leading to an increase in air and seawater temperature. As a result, an increase in the ocean temperature from 2 to 6 °C is predicted by the end of this century [5].

The effects of increased $CO_2$ and changes in pH and temperature have been studied on phytoplankton and microorganisms, in general, but often with contradictory results. Some reports speculate that marine phytoplankton may be resistant to ocean acidification [6–8], and could even stimulate cell abundance [9]. Others claim that small changes in pH can alter microbial structure [10], while warmer temperatures can increase phytoplankton abundance and microbial activity [5]. It is extremely important to understand how microbial communities will respond to climate change, especially those involved in nutrient cycles, such as $N_2$ fixing microorganisms, also called diazotrophs, which provide new inorganic nitrogen (N) to the oceans by reducing the atmospheric nitrogen ($N_2$) to inorganic nitrogen [11]. Several studies have described the role of $CO_2$ and temperature in diazotrophs, although these reports focus mainly on cyanobacteria, e.g., the filamentous *Trichodesmium*, or the unicellular *Cyanothece* and *Crocosphaera* [12–15]. In general, they show that the responses are dependent the availability of nutrients and light. Cyanobacteria can fix $CO_2$ and thus buffer the negative effects of $CO_2$ on ocean acidification. In some cyanobacterial species, elevated $CO_2$ can reduce the activity of carbon concentration mechanisms (CCMs) in carboxysomes, saving energy that can be used to increase $N_2$ fixation and diazotrophic growth [14]. However, different cyanobacterial species may have different CCMs, resulting in different $CO_2$ sensitivity, suggesting that the effect of elevated $CO_2$ levels in cyanobacteria is species-specific [16].

In recent years, the potential of heterotrophic bacteria as a source of new nitrogen has attracted attention. Several studies have reported the widespread distribution and expression of nitrogenase genes (*nifH*, *nifK*, and *nifD*) in heterotrophic bacteria, suggesting that they may play a very important role in global $N_2$ fixation rates [17,18]. Heterotrophic bacteria may be associated with marine particles, wherein their $N_2$ fixation activities can take place under low-oxygen conditions [19]. However, very little is known about the responses of heterotrophic $N_2$ fixers in the context of $CO_2$, acidification, and warming. Interestingly, some heterotrophic bacteria can fix and assimilate $CO_2$ through various carboxylation reactions via phosphoenolpyruvate carboxylase and pyruvate carboxylase, producing oxaloacetate, which can be taken up into the Krebs cycle [20]. Because cyanobacteria and heterotrophic bacteria use and assimilate $CO_2$ in different ways, they may respond differently to climate change, and their effects on nitrogen cycles are unknown.

It is predicted that $N_2$ fixation rates will increase to 22–27% by the end of this century due to an increase in water temperature [21–23]. These general predictions should be viewed with caution because the $N_2$ fixing community is composed of diverse groups that may differ in their optimal pH, temperature, and nutrient requirements for their growth and microbial activities [6,24–26]. These factors may regulate their responses to future ocean acidification and warming scenarios [27]. It has been well described that diazotrophic microorganisms have particularly high nutrient requirements, especially with respect to phosphorus (P) and iron (Fe), compared to other microorganisms [28,29], as P, i.e., an ATP source, and Fe, i.e., a cofactor of the nitrogenase complex, are required to catalyze the $N_2$ fixation reaction [30].

In our previous studies, we described how climate change ($CO_2$ and temperature) can alter $N_2$ fixation rates and the diazotrophic community structure on the endemic Mediterranean seagrass, *Posidonia oceanica* [31–34]. However, nothing is known about how diazotrophs associated with *P. oceanica* behave under these environmental conditions. Given the sensitivity of *P. oceanica*-associated cyanobacteria and heterotrophic bacteria to various abiotic factors, e.g., nutrients or emerging pollutants (microplastics) [35–38], it is of utmost importance to understand and predict how they individually respond to climate change factors based on their lifestyle. Some evidence shows that alkaline phosphatase (APA) activity plays a key role when cells are P-limited and contributes to $N_2$ fixation processes, thereby increasing the inorganic phosphorus pool of organic

phosphorus compounds [34,36]. Radical oxygen species (ROS) are biochemical biomarkers that accumulate when cells are exposed to environmental stress [35]. Therefore, the use of these parameters as biomarkers can contribute to the understanding of the effects of climate change factors on the diazotrophic population.

Considering the background information provided, it is expected than cyanobacteria and heterotrophic bacteria will respond differently to the climate change factors, and here we investigated the individual responses of four culturable diazotrophs (two cyanobacteria and two heterotrophic bacteria) associated with *Posidonia oceanica* to changes in (I) pH (from pH 4 to 8) and (II) different temperature levels (12–30 °C), both under different nutrient concentrations of inorganic phosphorus ($PO_4^{3-}$) and Fe. In addition, we focused on the unicellular cyanobacterium *Halothece* sp. and the heterotrophic bacterium *Cobetia* sp. to evaluate the effects of (III) different $CO_2$ concentrations (atmospheric $CO_2$, $aCO_2$: 410 ppm and elevated $CO_2$ as predicted for the end of the century, $eCO_2$: 1000 ppm) under different nutrient (P and Fe) and temperature (18–24 °C) levels. We evaluated the physiological responses in terms of growth and $N_2$ fixation activities. We also measured the P-acquisition mechanisms (APA), and oxidative stress (by measuring ROS). In contrast with previous studies focusing on cyanobacteria, we compared the responses based on lifestyle (cyanobacteria and heterotrophic bacteria), providing biomarkers for understanding the climate change effect on marine diazotrophs.

## 2. Materials and Methods

### 2.1. Culture Strains Tested

Based on our previous sequencing work with Illumina [38] and Oxford Nanopore (data did not shown), we selected four $N_2$ fixers that occur in association with *P. oceanica* and are represented by their most related culturable strains: two cyanobacteria, i.e., a unicellular *Halothece* sp. PCC 7418 and a filamentous heterocyst-forming *Fischerella muscicola* PCC 73103, and two heterotrophic bacteria, i.e., *Pseudomonas azotifigens* DSM 17556[T] and *Cobetia* sp. UIB 001, which was isolated from the roots of *P. oceanica* as an $N_2$ fixer [37] (Supplementary Table S1). Prior to the experiments, stock cultures were maintained in their optimal culture media: ASNIII + Turks Island Salts 4X for *Halothece* sp., $BG11_0$ for *F. muscicola* and marine broth for the heterotrophic bacteria. They were incubated at 24 °C and 120 rpm in a rotatory shaker with a photoperiod of 12 h dark:12 h light under low-intensity fluorescent light (30 $\mu$E m$^{-2}$ s$^{-1}$).

### 2.2. Experimental Culture Conditions

All experiments were performed in triplicate (n = 3) and in 2 mL microtiter plates (1.8 mL) or 50 mL falcon tubes (30 mL) in batch cultures using a modified artificial seawater medium [36]. Inorganic phosphorus (P, in the form of $K_2HPO_4$), iron (Fe, in the form of ferric citrate, [$C_6H_5FeO_7$]) and inorganic nitrogen (N, in the form of $NH_3Cl$) were added depending on the nutrient regimen and the response variable selected (Supplementary Table S1). Cells were precipitated and washed with artificial seawater without P, Fe and N, inoculating ~20–60 × $10^4$ cells mL$^{-1}$ in each treatment. In all experiments, cells were then maintained under 120 rpm, 12 h dark:12 h light and under low-intensity fluorescent light (30 $\mu$E m$^{-2}$ s$^{-1}$) for 72 h under the different experimental conditions tested (Supplementary Table S1). To avoid Fe contamination, all the samples and materials (acid-cleaned) were manipulated in a class-100 clean hood. After 72 cell abundance, when cell reached mid-exponential phase, $N_2$ fixation rates, alkaline phosphatase activity (APA), and reactive oxygen species (ROS) were measured as response variables (Supplementary Table S1).

#### 2.2.1. Effects of Variation in pH and Temperature Levels

The pH of the cultures was adjusted to different levels—pH 4, 5, 6, 7 and 8—using Tris-HCl or Trizma base (Sigma-Aldrich, Burlington, MA, USA) (0.1 mM final concentration) in a calibrated pH-Meter BASIC 20 (CRISON, Barcelona, Spain) (Supplementary Table S1). These values were chosen based on the microenvironments found in seagrasses, where pH

can be less than 6 [39]. Cells were incubated at a constant temperature (24 °C), which is the temperature at which all bacterial species tested exhibited high growth rates. For the temperature experiment, cultures were incubated at different temperatures levels—12, 18, 24 and 30 °C—using coolers and thermostats, corresponding to sea surface temperature in the Mediterranean Sea (Supplementary Table S1). The initial pH was set at 8, and cultures were incubated at different nutrient concentrations in both pH and temperature experiments: at optimal nutrient concentrations (1.5 mM $PO_4^{3-}$ and 1 μM Fe), based on the optimal P and Fe concentration of the artificial seawater medium, and under nutrient limitation (0.1 μM $PO_4^{3-}$ and 2 nM Fe), based on our previous studies on nutrient limitation in these bacteria [34–37], both under low levels of $NH_3$ (0.15 mM) to avoid $N_2$ fixation inhibition [40].

### 2.2.2. Effects of Increased CO$_2$ Levels

To investigate the effects of increased $CO_2$, we chose the unicellular cyanobacterium, *Halothece* sp., and the heterotrophic bacterium, *Cobetia* sp. Because of the complexity of the experimental design due to the large number of combined variables we focused only on these bacteria as models, which were already selected in our previous studies [34–37]. We tested two different $CO_2$ concentrations: atmospheric, a$CO_2$: 410 ppm; and elevated, e$CO_2$: 1000 ppm (as predicted for the end of the century, IPCC, 2014), and included a control without $CO_2$ supply. All experiments were performed under low $NH_3$ concentrations (0.15 mM). Combinations with two variables (Fe and temperature) and different $CO_2$ levels were conducted: (i) $CO_2$ and Fe (1 μM and 2 nM) at constant temperature (24 °C); and (ii) $CO_2$ at temperatures of 18 °C and 30 °C with 1 μM of Fe (Supplementary Table S1). Both experiments were subject to different concentrations of $PO_4^{3-}$ (optimal, 1.5 mM; limited, 0.1 μM), and the initial pH was adjusted to 8 in each treatment.

The microtiter plates and Falcon tubes containing the cultures were placed in hermetically sealed tanks without bubbling (Supplementary Figure S1). Mass flow controllers (MFCs, Aalborg, Denmark) were used to control the air mixture. The gas mixture was introduced into the hermetic tanks with an outlet port allowing the gas mixture to flow through. To achieve these mixtures, air from an air pump was connected to a filter with soda lime to remove $CO_2$ and mixed with pure $CO_2$ from a bottle to achieve the desired concentration. The mixing was done in a tube with marbles to ensure homogenization of the gasses. After splitting the resulting treatment air–gas mixture, the volume entering each tank was controlled by a flow meter with a volume of 2.5 L min$^{-1}$ (LPM). The pH was continuously measured and monitored every 30 min (ENV-40-pH), and the resulting data were stored in two control boxes (IKS-AQUASTAR) (Supplementary Figure S1).

### 2.3. *Flow Cytometry and Growth Measurement*

Fresh unfixed cells at the initial ($T_o$) and the final time point ($T_f$) of the experiments were counted using a Becton Dickinson FACS-Verse cytometer (Beckton and Dickinson, Franklin Lakes, NJ, USA). Fluorescent beads, BD FACSuite™ CS&T research beads (Beckton and Dickinson and Company BD Biosciences, San Jose, CA, USA), were used as an internal standard to calibrate the instrument. Cells were separated by combinations of the flow cytometer parameters: forward scatter (FSC, reflecting cell size), side scatter (SSC, reflecting cell internal complexity) and/or fluorescein isothiocyanate (FITC, 488 nm excitation, 530/30 nm emission) parameters, with a total of $1 \times 10^4$ cells recorded for each sample.

### 2.4. *Nitrogen Fixation Activity*

$N_2$ fixation rates were measured at the end of the experiments after 72 h using the acetylene reduction assay (ARA) according to the general method described in [41]. Only cultures cultured under P-optimal conditions were measured. Briefly, a known volume of culture media (8 mL) was transferred to a 10 mL vial. Liquid saturated acetylene was injected into these vials, reaching a final concentration of 20% (*v/v*), and incubated for 3 h. Ethylene and acetylene were determined using a gas chromatograph (model HP-5890,

Agilent Technologies, Santa Clara, CA, USA) equipped with a flame ionization detector (FID). The column was a Varian wide-bore column (ref. CP7584) packed with CP-PoraPLOT U (27.5 m length, 0.53 mm inside diameter, 0.70 mm outside diameter, 20 μm film thickness), using the set up described in [34–37]. Ethylene produced was calculated using the equations in [42]. The acetylene reduction rates were converted to $N_2$ fixation rates (nmol mL$^{-1}$ h$^{-1}$) by a factor of 4:1 [43].

### 2.5. Determination of the Alkaline Phosphatase Activity

Alkaline phosphatase activity (APA) was assessed after 72 h by a fluorometric assay measuring the hydrolysis of the substrate 4-methylumbelliferyl phosphate (MUF-P, Sigma-Aldrich, Burlington, MA, USA) to 4-methylumbelliferyl (MUF) [35]. An enzymatic endpoint assay was performed at the end of the experiment using 8 μM of MUF-P (Supplementary Table S1). Only cultures cultured under P-limiting conditions were measured. After 1 h incubation in the dark, APA was measured in a microtiter plate containing borate pH buffer 10 (3:1 of sample: buffer). MUF production was measured at 359 nm (excitation) and 449 nm (emission) using a calibration standard curve with commercial MUF (Sigma-Aldrich), with the Cary Eclipse spectrofluorometer (FL0902M009, Agilent Technologies, Santa Clara, CA, USA).

### 2.6. Determination of Reactive Oxygen Species

The molecular probe 2′-7′-dichlorofluorescein diacetate (DCFH) (Sigma-Aldrich, Burlington, MA, USA) was used to measure reactive oxygen species (ROS) production after 72 h [34]. Only cultures cultured under P-optimal conditions were measured. DCFH diluted in artificial seawater (previously diluted in acetone to 1 mg mL$^{-1}$) was added to a 96-well microplate (Thermo Scientific, Waltham, MA, USA) containing the bacterial samples, reaching a final concentration of 15 μg mL$^{-1}$. Emission of green fluorescence from the resulting 2′-7′-dichlorofluorescein was measured using a Cary Eclipse spectrofluorometer (FL0902M009, Agilent Technologies). Fluorescence was monitored for 1 h with excitation at 480 nm and emission at 530 nm. The slope of the linear regression between fluorescence and elapsed time was expressed as ROS production expressed in arbitrary units (A.U.) and normalized to cell number. DCFH was added to artificial seawater without cells under the same conditions as above and served as a blank.

### 2.7. Statistical Analysis

Normality and homogeneity of the results obtained in the pH, temperature and $CO_2$ experiments were checked, and the statistical significance level was set at $p < 0.05$. Levene test shown the homogeneity of the data, and parametric analysis was used to examine normally distributed data using ANOVAs with Bonferroni post hoc test. We applied the same statistical analyses for all the experimental designs. Spearman correlation analysis was used to determine the relationships between ROS vs. cell abundance, and APA vs. cell abundance. All analyses were performed in R-Studio, R version 3.6.3 (29 February 2020).

## 3. Results

### 3.1. Impact of Climate Change on Growth Responses

#### 3.1.1. Effect of pH and Temperature

The growth responses of the tested diazotrophs to different pH and temperature levels were dependent on the nutrient status of the cells (Figures 1 and 2; $p < 0.05$). For all pH and temperature treatments, cells under nutrient-rich conditions (P and Fe) grew 4-fold more than cells under nutrient-poor conditions (Figures 1A and 2). The extent of the difference between growth under nutrient-rich cells and nutrient-poor cells varied depending on the species and pH studied (Figure 1A,B). Under nutrient-rich conditions, a significant linear regression was found in which decreasing pH negatively affected growth of all species tested (Figure 1A, Spearman correlation; $p < 0.05$, n = 20, $r^2 = 0.6$). Under these conditions, we did not detect any differences between cyanobacteria or heterotrophic bacteria, as the cyanobacterium *F. muscicola* and the heterotrophic bacterium *Cobetia* sp.

were the species most affected by the low pH, as both reduced their growth by 30-fold (Figure 1B; $p < 0.05$). Under nutrient limiting conditions, we found the opposite response and reported a negative regression (non-linear), where increasing pH reduced growth (Figure 1B, Spearman correlation; $p < 0.05$, n = 20, $r^2 = -0.2$). Different responses were observed between cyanobacteria and heterotrophic bacteria under these conditions. The optimal pH for growth of the tested cyanobacterial species shifted towards an acidic pH (pH 5), with a 9-fold increase in growth compared to the other pH treatments (Figure 1B; $p < 0.05$). A slight change from pH 8 to an acidic neutral pH (at pH 6–7) was also observed in the heterotrophic species tested under nutrient limitation (Figure 1B).

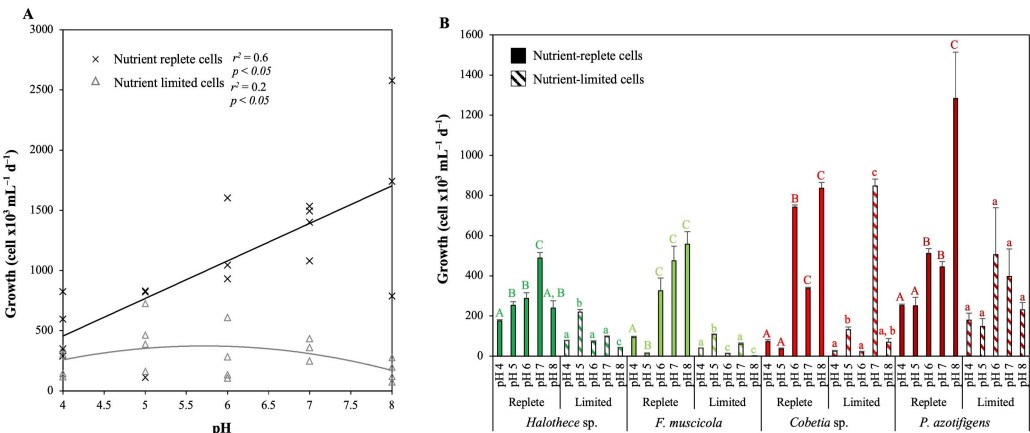

**Figure 1.** Effect of different levels of pH (4–6) on growth after 72 h under different nutrient concentrations at 24 °C. (**A**) Regression analysis between growth (cell $\times 10^3$ mL$^{-1}$ d$^{-1}$) and pH level according to the nutritional level (replete or limited) in all the strains. (**B**) Effect of pH (4–8) according to the species tested and nutritional level on growth. In (**B**), values are the mean $\pm$ SE (n = 3). Letters (i.e., capital letters for the nutrient-replete condition or lowercase for the nutrient-limiting condition) indicate significant differences ($p < 0.05$) with the rest of the treatments for each strain and nutrient level, using a post hoc test (Bonferroni's test) after ANOVA over the whole dataset.

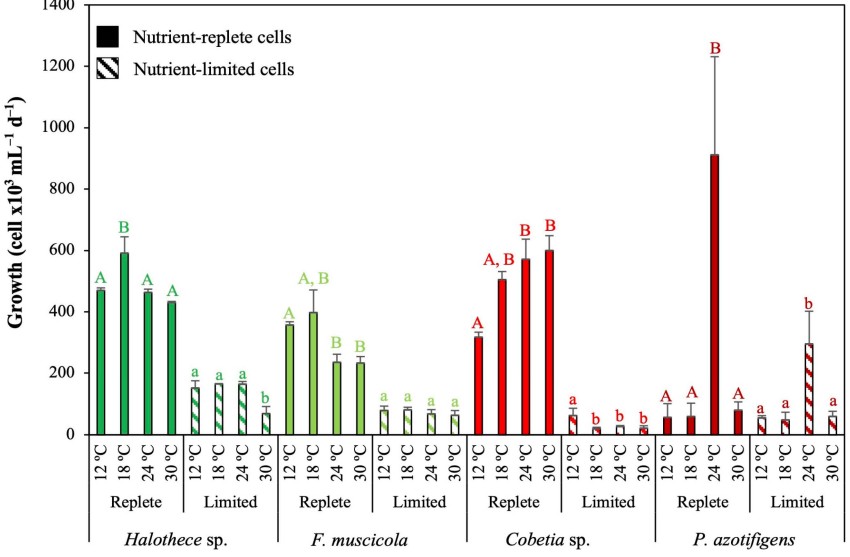

**Figure 2.** Temperature effect on growth after 72 h under different nutrient concentrations. Values are the mean $\pm$ SE (n = 3). Letters (i.e., capital letters or lower case) indicate significant differences ($p < 0.05$) with the rest of the treatments for each strain and nutrient level, using a post hoc test (Bonferroni's test) after ANOVA over the whole dataset.

The response of the $N_2$ fixers tested at different temperatures was also dependent on the nutrient status of the cells and the species (Figure 2; $p < 0.05$). Among heterotrophic bacteria, the nutrient-rich cells, *Cobetia* sp. and *P. azotifigens*, grew better at warmer temperatures (24–30 °C) than the cyanobacterial strains, *Halothece* sp. and *F. muscicola* (12–18 °C) (Figure 2). However, under nutrient limiting conditions, the optimal growth temperature of *Cobetia* sp. was 12 °C (Figure 2; $p < 0.05$), while for *P. azotifigens*, we did not detect significant changes under nutrient limitation (Figure 2; $p > 0.05$). Under the same nutrient conditions for the cyanobacterial strains, their optimal temperature range increased up to 24 °C (Figure 2; $p < 0.05$).

### 3.1.2. Effect of $CO_2$

Elevated $CO_2$ (e$CO_2$, 1000 ppm) resulted in lower pH than at ambient levels (a$CO_2$, 410 ppm) after 72 h (Supplementary Figure S2). The pH differences between the two $CO_2$ treatments were lower for the cyanobacterium, *Halothece* sp. (an average of 0.21 units of pH), than for the heterotrophic bacterium, *Cobetia* sp. (an average of 0.55 units of pH). The growth of the unicellular cyanobacterium tested (*Halothece* sp.) did not vary significantly at the different $CO_2$ levels applied in our study under different nutrient regimes (P and/or Fe) nor temperature levels (18 °C and 30 °C) (Figure 3A–D, $p > 0.05$).

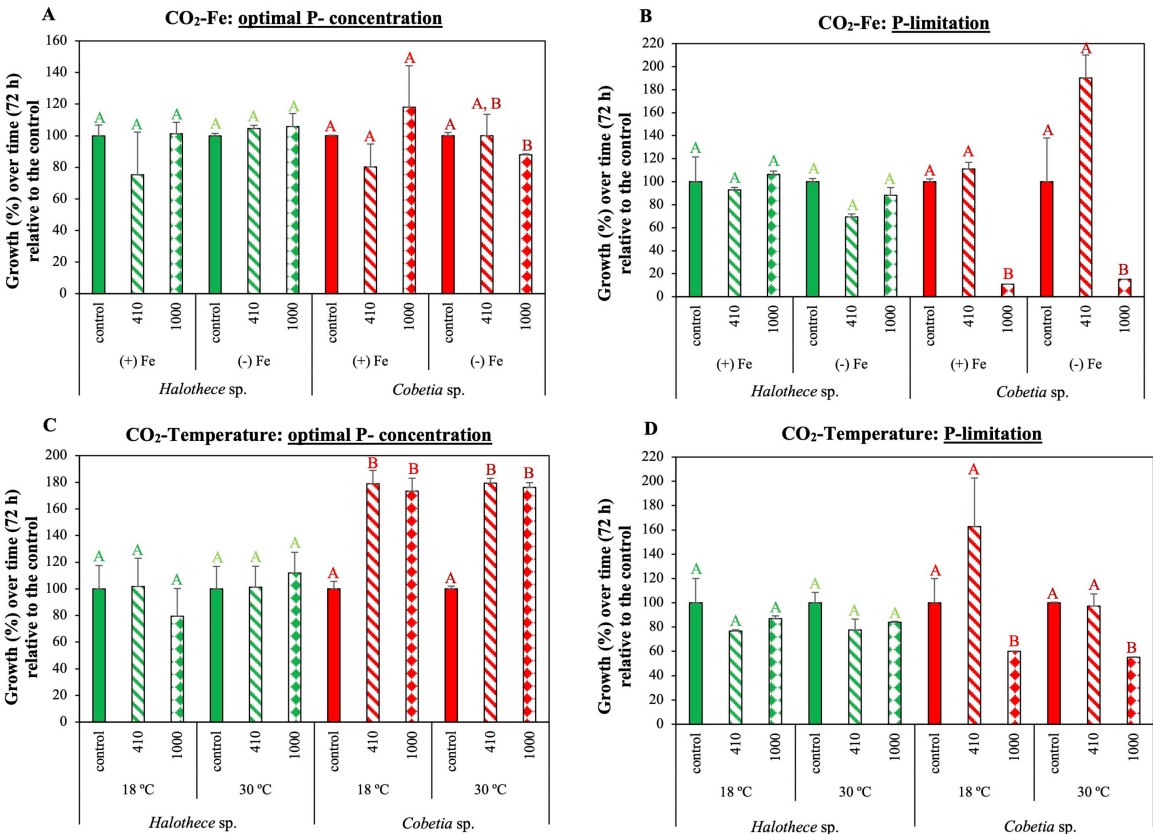

**Figure 3.** Growth (calculated as % difference relative to the control over the incubation time of 72 h) after the exposure to atmospheric $CO_2$ (a$CO_2$, 410 ppm) and elevated $CO_2$ (e$CO_2$, 1000 ppm). Effect of $CO_2$ exposure on growth at different concentrations of Fe (+Fe, 1 μM and −Fe, 2 nM) under (**A**) P-optimal conditions (1.5 mM) and (**B**) P-limitation (0.1 μM). Exposure to $CO_2$ at different temperature levels (18 °C and 30 °C), under (**C**) P-optimal conditions, and (**D**) P- limitation. In (**C**,**D**), cultures were incubated under 1 μM of Fe. Values are the mean ± SE ($n = 3$). Letters indicate significant differences ($p < 0.05$) with the rest of the treatments for each strain and experimental set-up, using a post hoc test (Bonferroni's test) after ANOVA over the whole dataset.

Conversely, for the heterotrophic bacterium, *Cobetia* sp., different $CO_2$ concentrations combined with varying nutrient concentrations had a significant effect on its growth (Figure 3A–D; $p < 0.05$). In both experiments, $CO_2$-Fe and $CO_2$-temperature, growth at $eCO_2$ was lower compared with $aCO_2$ only when cultures were nutrient-limited (Figure 3B,D; $p < 0.05$). In the $CO_2$-Fe experiments, growth was significantly lower at $eCO_2$ compared to the control (no $CO_2$ influx) when cells were growing without Fe at optimal P concentration (Figure 3A; $p < 0.05$). However, P was the limiting factor, as under P limitation, growth was dramatically reduced in $eCO_2$ compared to control, regardless of Fe level (Figure 3A,B; $p < 0.05$). In the $CO_2$-temperature experiments, under P-optimal concentrations, growth was higher at $aCO_2$ and $eCO_2$ compared to the control, irrespective of the temperature level (Figure 3C; $p < 0.05$), while under P-limited conditions, growth at $eCO_2$ was lower compared to the control, irrespective of the temperature level (Figure 3C,D; $p < 0.05$). Altogether these results show the importance of nutrient regimes in modulating climate change factors on diazotrophic growth responses.

### 3.2. Effect of Climate Change on $N_2$ Fixation Rates
#### 3.2.1. Effect of pH and Temperature

Although both groups had higher $N_2$ fixation rates at alkaline pH (Figure 4A), heterotrophic bacteria maintained active $N_2$ fixation rates at pH 6–8 (Figure 4A), while cyanobacteria have active $N_2$ fixation rates under acidic conditions (<pH 7) (Figure 4A). Regarding temperature, cyanobacteria, i.e., *Halothece* sp. and *F. muscicola*, reached the maximum $N_2$ fixation rates at 24 °C, while in the heterotrophic bacteria, it was at 18 °C (Figure 4B). Moreover, our results indicate that temperatures over 30 °C may have an inhibitory effect on the $N_2$ fixation processes (Figure 4B). However, these results were not statistically significant.

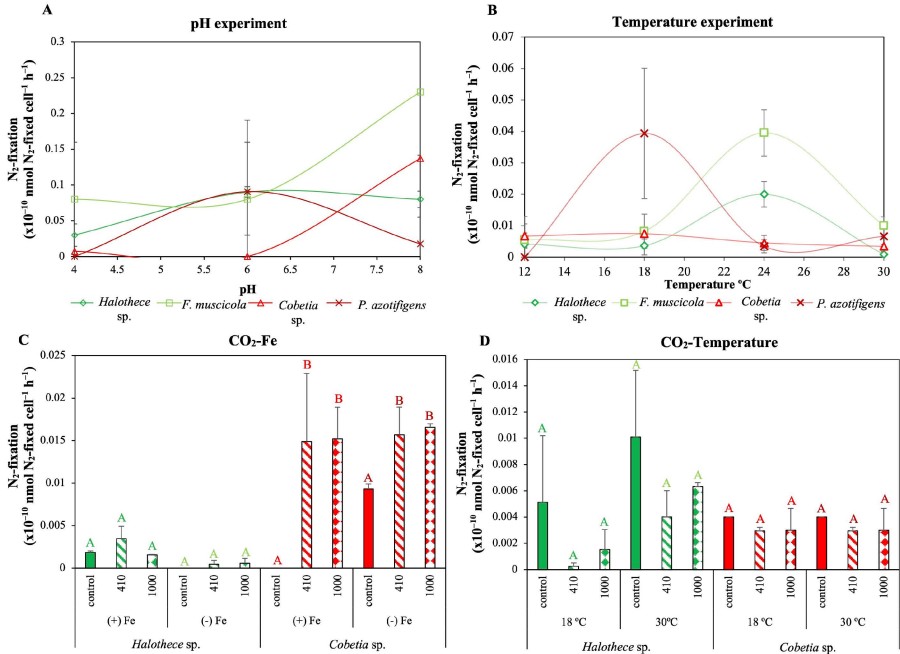

**Figure 4.** Effect of global climate change factors on $N_2$-fixation rates. Effect of (**A**) pH (4–8) and (**B**) temperature levels (12–30 °C) on $N_2$ fixation rates after 72 h (as line plots). (**C,D**) Effect of different concentrations of $CO_2$ (control, atmospheric $CO_2$ ($aCO_2$): 410 ppm, and elevated $CO_2$ ($eCO_2$): 1000 ppm) in combination with different (**C**) Fe (+Fe, 1 µM and −Fe, 2 nM) and (**D**) temperature levels (18 °C and 30 °C). All treatments were incubated under optimal P-concentration to detect $N_2$-fixation rates. Values are the mean ± SE (n = 3). Letters indicate significant differences ($p < 0.05$) with the rest of the treatments for each strain and experimental set-up, using a post hoc test (Bonferroni's test) after ANOVA over the whole dataset.

### 3.2.2. Effect of $CO_2$

No significant changes in the $N_2$ fixation rates, irrespective of the $CO_2$, Fe and/or temperature levels were found for *Halothece* sp. (Figure 4C,D; $p > 0.05$). For the heterotrophic species tested, *Cobetia* sp., continuous influx of $CO_2$, both $aCO_2$ and $eCO_2$ significantly enhanced the $N_2$ fixation rates compared to the control in the $CO_2$-Fe experiments at 24 °C, irrespective of the Fe level (Figure 4C; $p < 0.05$).

### 3.3. Plasticity of Diazotrophs against Climate Change: Changes in P-Acquisition Mechanisms and Oxidative Stress

#### 3.3.1. Changes in P-Acquisition Mechanisms

Ocean acidification and higher temperatures could impair alkaline phosphatase activity (APA) (Figure 5A,B). For *Halothece* sp. and *Cobetia* sp., the maximum APA was achieved at alkaline pH (above pH 7) (Figure 5A), and the rates dropped significantly from pH 8 to 6.5 for both bacteria (Figure 5A; $p < 0.05$). Different temperature levels also affected the APA in the diazotrophs tested (Figure 5B; $p < 0.05$). The optimal temperature for the APA varied by species (for cyanobacteria at 24 °C and heterotrophic bacteria at 12–18 °C). However, for all the species we tested, APA had lower rates at 30 °C (Figure 5B; $p < 0.05$), and some strains, e.g., *Halothece* sp. and *P. azotifigens*, showed high activities at cold temperatures.

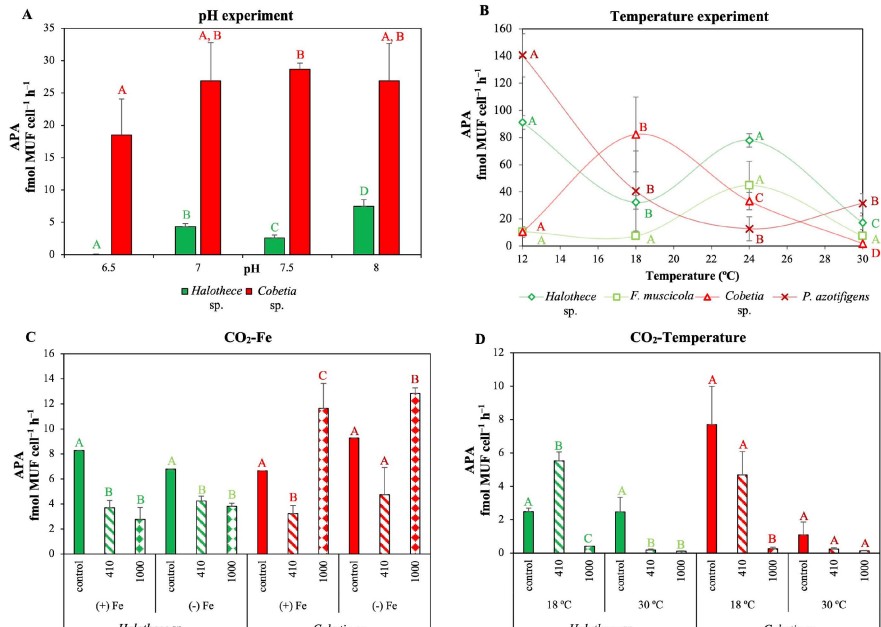

**Figure 5.** Effect on P-acquisition mechanisms (alkaline phosphatase activity [APA]), after 72 h, under different (**A**) pH (6.5–8) and (**B**) temperature levels (12–30 °C) as a line plot, and (**C,D**) different concentrations of $CO_2$ (control, atmospheric $CO_2$ ($aCO_2$): 410 ppm, and elevated $CO_2$ ($eCO_2$): 1000 ppm) in combination with different (**C**) Fe (+Fe, 1 μM and −Fe, 2 nM) and (**D**) temperature levels (18 °C and 30 °C). The measurements were taken under P limitation (0.1 μM) to induce APA. Note that for the pH experiment, APA was only measured in *Halothece* sp. and *Cobetia* sp. in a pH range between 6.5–8. Values are the mean ± SE (n = 3). Letters indicate significant differences ($p < 0.05$) with the rest of the treatments for each strain and experimental set-up, using a post hoc test (Bonferroni's test) after ANOVA over the whole dataset.

For both strains, the $CO_2$ supply triggered changes in the APA, and the temperature was the main factor regulating the responses (Figure 5C,D). Generally, for *Halothece* sp., under both, $aCO_2$ and $eCO_2$ levels, APA decreased significantly compared to the control treatment (Figure 5C,D; $p < 0.05$). However, we did not find any significant differences between $aCO_2$ and $eCO_2$ (Figure 5C,D; $p > 0.05$). APA was negatively affected by increasing temperatures (especially at 30 °C) (Figure 5C; $p < 0.05$), as we previously reported in the

temperature experiments (Figure 5B). For the heterotrophic species, *Cobetia* sp., elevated $CO_2$ levels enhanced the APA irrespective of the Fe levels at 24 °C compared with the control and compared with a$CO_2$ levels in the Fe-$CO_2$ experiments (Figure 5C; $p < 0.05$). Yet, elevated $CO_2$ levels significantly reduced APA at 18 °C, compared to the control and a$CO_2$ levels in the $CO_2$-temperature experiments (Figure 5D; $p < 0.05$). At 30 °C, we did not find any significant differences among the $CO_2$ treatments (Figure 5D; $p > 0.05$).

### 3.3.2. Changes in ROS

Significantly high ROS production was measured for *Halothece* sp., *F. muscicola* and *P. azotifigens* at the pH of normal ocean water (pH 8) compared to other pH treatments. However, we found species-specific responses since for *Cobetia* sp. pH 8 decreased ROS (Figure 6A; $p < 0.05$). Temperature also affected ROS production, but their response was also dependent on the species tested (Figure 6B; $p < 0.05$). For *Halothece* sp. and *P. azotifigens*, ROS production was lower at intermediate temperatures than at the lowest (12 °C) and highest (30 °C) temperature treatments (Figure 6B; $p < 0.05$). For the other species, i.e., *F. muscicola* and *Cobetia* sp., no significant differences were detected (Figure 6B; $p > 0.05$).

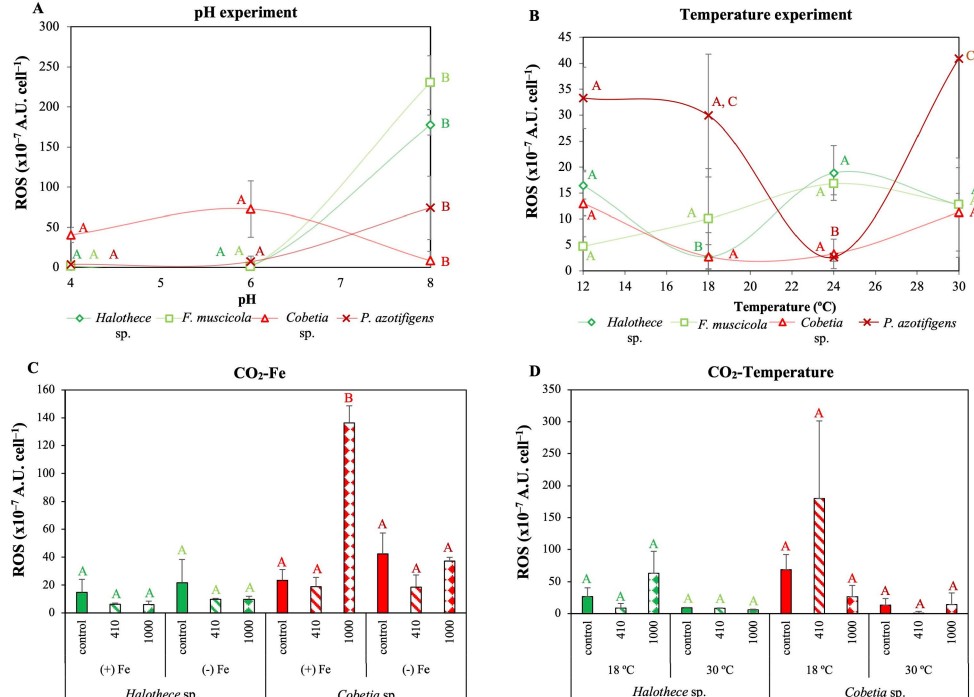

**Figure 6.** Reactive oxygen species (ROS) production after 72 h, under different (**A**) pH and (**B**) temperature levels, and (**C,D**) different concentrations of $CO_2$ (control, atmospheric $CO_2$ (a$CO_2$): 410 ppm, and elevated $CO_2$ (e$CO_2$): 1000 ppm) in combination with different (**C**) Fe (+Fe, 1 μM and −Fe, 2 nM) and (**D**) temperature levels (18 °C and 30 °C). All treatments were incubated under optimal P-condition. Values are the mean ± SE (n = 3). Letters indicate significant differences ($p < 0.05$) with the rest of the treatments for each strain and experimental set-up, using a post hoc test (Bonferroni's test) after ANOVA over the whole dataset.

ROS production was not significantly affected in response to the addition of $CO_2$ combined with different Fe and temperature levels for *Halothece* sp. (Figure 6C,D; $p > 0.05$). Conversely, for the heterotrophic bacteria *Cobetia* sp., e$CO_2$ levels with added Fe, ROS production increased 5-fold compared with the control and a$CO_2$ (Figure 6C; $p < 0.05$). However, no significant ROS production was observed with the addition of $CO_2$ at different temperature levels (Figure 6D; $p > 0.05$).

### 3.3.3. APA and ROS as Molecular Biomarkers

Cell abundance of all the species and treatment tested (pH, temperature, $CO_2$, and nutrient level) were negatively correlated with the cell specific production of APA and ROS (Figure 7A,B, Spearman's correlation; $p < 0.05$, n = 144, $r^2 = -0.34$ and $r^2 = -0.41$, respectively).

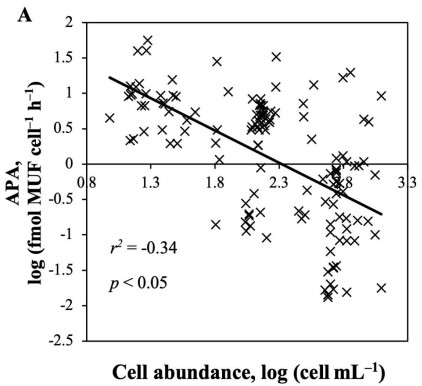
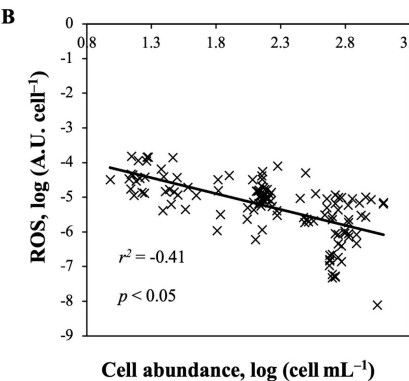

**Figure 7.** Linear regression analyses between (**A**) cell abundance and ROS per cell and (**B**) cell abundance and APA per cell. All treatments and species from the pH, temperature, and $CO_2$ experiments were combined.

## 4. Discussion

### 4.1. Dependence of Nutrient Status of $N_2$ Fixers in Their Growth Responses to Climate Change Factors

We demonstrate that P and Fe concentrations can control the growth responses of $N_2$ fixers to climate change factors (pH, temperature, and $CO_2$), but that their response also depends on their nature (cyanobacteria or heterotrophic bacteria) (Figures 1–3). Diazotrophic bacteria are primarily controlled by the availability of P and Fe [11]. Therefore, diazotrophic bacteria require large amounts of P to drive $N_2$ fixation, requiring up to 16 ATP [44]. Iron (Fe) is also a primary factor limiting $N_2$ fixation, as it is a co-factor of the nitrogenase complex [45], and it is particularly critical for cyanobacteria as Fe can limit photosynthetic processes [46]. Nutrient-limited conditions can promote higher alkaline phosphatase activity (APA), oxidative stress, cell breakage, and apoptotic processes, as reported in our previous studies [35,36,38]. This explains the higher growth in nutrient-rich cells compared to a nutrient-limiting situation (Figures 1 and 2). We found no difference between these microbial groups (Figures 1 and 2). However, cyanobacteria and heterotrophic bacteria may have different nutrient requirements as cyanobacteria have higher nutrient requirements than other phytoplankton taxa, as suggested by their Redfield N:P ratio of 25, which is significantly higher than other marine microbial groups with a ratio of 16. These data indicate that P and Fe may be the main abiotic factors controlling the responses of these $N_2$ fixers.

Although several reports have focused on the effects of pH in soil $N_2$ fixers [47–49], to the best of our knowledge, this is the first study to examine the independent effects of ocean acidification on marine diazotrophs independent of $CO_2$ level. Previous studies suggest that marine microorganisms are not affected by small changes in pH [7]. We also found that the species tested were not affected by pH changes dropping to 6.5 (data not shown). Therefore, future pH changes in the water column, dropping the pH to 7.6–7.7, should not affect the diazotrophs, considering that it is improbable that the oceans will reach an acidic pH (i.e., below pH 7). However, seagrasses are complex systems. The epiphytic population in the rhizosphere can be subjected to pH microenvironments, finding very low pH values [39]. Ocean acidification can exacerbate this process, decreasing the pH levels. Interestingly, growth responses below pH 6 are dependent on nutritional status and bacterial nature, i.e., cyanobacteria or heterotrophic bacteria (Figure 1A,B). Under nutrient-replete conditions, it has already been described in phytoplankton that decreasing pH levels

negatively affect cell growth [6] (Figure 1A,B). Lower pH values decrease Fe uptake because Fe (III) is less available to organic chelators, e.g., siderophores, and Fe (II). The reduction in Fe uptake along with changes in cell membranes and gene expression may limit microbial activity at low pH under rich nutrient conditions [12,50–52]. Surprisingly, we found that the pH optimum shifted toward a more acidic pH under nutrient-limited conditions (Figure 1A), especially for cyanobacterial species (*Halothece* sp. and *F. muscicola*), as these species grow better below pH 6 (Figure 1B). These results indicate that cyanobacteria can be more resistant to lower pH than heterotrophic bacteria (*Cobetia* sp.) (Figure 1B). How these species profit from an acidic pH (high [$H^+$]) under conditions of nutrient limitation is unknown, although some cyanobacteria have a high affinity for $Na^+/H^+$ antiporters, through which intracellular $Na^+$ is exported to maintain cell homeostasis [53]. These data highlight the differential survival strategies that phototrophic and heterotrophic bacteria exhibit against pH changes in marine environments.

Temperature is a key factor in microbial activity in the oceans and plays a crucial role in bacterial metabolism [54]. In a nutrient-rich situation, heterotrophic bacteria prefer warmer temperatures ($\geq$24 °C) than cyanobacteria ($\leq$18 °C) (Figure 2). Higher temperatures can stimulate respiration and growth of heterotrophic bacteria [54], whereas in cyanobacteria, warmer temperatures can reduce photosynthetic processes [55]. However, in some bacteria, e.g., *Cobetia* sp., we found the opposite growth response under nutrient-limiting conditions, where an increase in temperature reduced growth (Figure 2). This indicates that nutrient availability may be one of the main factors controlling the bacterial optimal temperature [54] (Figure 2). Early studies reported changes in optimal temperature from 3 to 6 °C, depending on nutrient status in different phytoplankton groups, e.g., *Synechococcus* sp., *Skeletonema costatum*, *Emiliania huxleyi* or *Thalassiosira pseudonana*, probably due to alterations in enzyme kinetics [56,57]. For heterotrophic bacteria, ocean warming can also increase nutrient consumption rates, nutrient limiting the media and liming growth [58]. Despite not addressing the effect of temperature on changes in shape and size here, it is well described that ocean warming reduces the bacterial size for metabolic advantages and may change the bacterial responses to face the environmental factors [59]. These results highlight again the species-specific responses that can be found between cyanobacteria and heterotrophic bacteria.

Cyanobacteria perform photosynthesis as phototrophs and remove $CO_2$ from the medium through efficient carbon concentration mechanisms (CCMs), that transport and concentrate $CO_2$ into the carboxysomes (structures encapsulating the $CO_2$-fixing enzyme RuBisCO) [60]. Thus, cyanobacteria can buffer the negative effect of $CO_2$, decreasing the pH level (Supplementary Figure S2). This explains why the cyanobacterium tested did not respond to elevated $CO_2$ levels, regardless of nutrient and temperature levels (Figure 3A–D). Our results are consistent with other studies on known cultures of $N_2$ fixing cyanobacteria (*Cyanothece* sp. ATCC51142, *Crocosphaera watsonii* WH850/WH0401/WH0402 and *Trichodesmium erythraeum* IMS101) [16,61–65]. However, in other studies on $N_2$ fixing cyanobacteria, e.g., *Crocosphaera*, *Cyanothece*, *Trichodesmium* and *Nodularia*, the authors report a growth increase or decrease due to elevated $CO_2$ levels depending on the nutritional status [64,66–68]. These data suggest different CCM efficiencies, and thus, the responses to climate change may be species-specific in cyanobacteria.

Conversely, a higher sensitivity against ocean acidification in heterotrophic bacteria was demonstrated, as they do not have efficient mechanisms to remove $CO_2$ from the media, and thus, they are more exposed to the acidification of the media (Supplementary Figure S2). Studies on $CO_2$ and heterotrophic $N_2$-fixing bacteria are scarce, and only a few evaluate the role of $CO_2$ in marine non-diazotrophic heterotrophic bacteria [69–71]. In agreement with these studies, elevated $CO_2$ levels in heterotrophic bacteria cultured under enriched nutrient media (P and Fe) do not affect microbial growth (Figure 3A,C). However, higher $CO_2$ concentrations could increase bacterial production, triggering changes in the metabolism [69,70]. Here, we report the negative effect of increasing $CO_2$ levels on heterotrophic growth only when cells are under low nutrient concentration, regardless of the temperature level (Figure 3A–D). We hypothesize that since low nutrient availability

can limit growth, protein synthesis and activity [72], a combined effect with decreasing pH due to increasing $CO_2$ levels could exacerbate the negative growth responses in heterotrophic bacteria. These results show the role of nutrients in controlling $CO_2$ responses in heterotrophic bacteria.

### 4.2. Climate Change Effect on $N_2$ Fixation

Higher pH may increase $N_2$ fixation rates (Figure 4A), which is consistent with the results of previous studies on soil samples. It is suggested that low cytosolic pH may alter the structure and catalysis of the nitrogenase complex [48,49]. Temperature changes affect cyanobacteria and heterotrophic bacteria differently, suggesting species-specific responses, but in both groups, a temperature of 30 °C had an inhibitory effect on $N_2$ fixation rates (Figure 4B). Some studies show that $N_2$ fixation rates are active within a temperature range of 24 to 30 °C, but activity decreases at temperatures > 30 °C, suggesting that temperature may control nitrogenase activity and expression [13,23]. Considering the effects of pH, temperature, and $CO_2$ on the growth of the tested diazotrophs (Figures 1 and 2), it is important to note that changes in their abundance may affect $N_2$ fixation rates.

In contrast to previous reports on other cyanobacterial species where $CO_2$ can increase or decrease $N_2$ fixation rates depending on nutrient and temperature [12], we did not detect significant changes in the cyanobacterium studied (*Halothece* sp.) (Figure 4C,D). In some cyanobacteria, high $CO_2$ can downregulate carboxysome genes, reducing the energy used for $CO_2$ concentration in these organelles [73]. It is hereby suggested that under high $CO_2$ levels, cyanobacteria can save energy by concentrating $CO_2$ and use this energy instead for $N_2$ fixation [14,74]. However, the absence of effects due to a $CO_2$ increase in the cyanobacterium studied suggests that the regulatory processes of carbon concentration or the mechanisms of $CO_2$ uptake may be species specific. Thus, the responses of diazotrophic cyanobacteria to $N_2$ fixation cannot be generalized when considering climate change. On the other hand, some of the results obtained for *Cobetia* sp. suggest that $CO_2$ may regulate $N_2$ fixation processes in heterotrophic bacteria (Figure 4C). Some heterotrophic bacteria can fix $CO_2$ by various carboxylation reactions catalyzed by phosphoenolpyruvate (PEP) carboxylase and pyruvate carboxylase. These enzymes can convert $HCO_3^-$ to oxalacetate, which can be included in the Krebs cycles [20] and contributes to 2–8% of the carbon content of cell biomass [75]. Therefore, some heterotrophic bacteria can exploit elevated $CO_2$ to increase the ATP pool and drive $N_2$ fixation processes. In the genome of *Cobetia* sp. (UIB 001) (accession number: CP058244-CP058245), we found that this bacterium contains a PEP carboxylase (locus tag: HA399_04700) and two SulP family inorganic anion transporters (locus_tag: HA399_00940 and HA399_13445), responsible for bicarbonate transport into the cells. These findings suggest its potential as $CO_2$-fixing bacteria (Figure 4C).

### 4.3. P-Acquisition Mechanisms and Oxidative Stress as Biomarkers for Predicting Microbial Perturbations Associated with Climate Change

#### 4.3.1. P-Acquisition Mechanisms

APA is activated as a survival mechanism when P is limited, or when cellular P demand increases. Thus, inorganic P is released from dissolved organic phosphorus (DOP), increasing the inorganic P pools in marine environments [76]. By measuring APA, it is possible to assess whether cells are P-limited, or they have fulfilled their P requirements (Figure 5A–D). The abiotic factors that $N_2$ fixers must cope with, such as climate change, could alter P-requirements and energy demand. It is already described the link between $N_2$ fixation and APA in *P. oceanica* [33]. Although lower APA may indicate that cells can have their P-requirements fulfilled, our results show a clear negative correlation between APA per cell and cell abundance (Figure 7A), suggesting that climate change factors may inhibit APA, and decrease the ability of cells to obtain inorganic P from DOP, thereby affecting the survival of $N_2$ fixers. Overall, we suggest that APA can be used as a biomarker to track changes in microbial community plasticity, as we describe below.

In general, acidic pH decreased APA rates in the cyanobacteria and heterotrophic bacteria tested (Figure 5A,C,D). Studies on cyanobacteria, e.g., *Nostoc flagelliforme*, and heterotrophic bacteria, e.g., *Cobetia amphilecti* KMM 296, show the dramatical inhibition of APA with acidification, possibly due to loss of enzyme stability [77–79]. Interestingly, APA was more sensitive to the cyanobacterial strain and was strongly inhibited below pH 7, whereas the heterotrophic bacterium was able to maintain higher rates of APA in the range of 6.5 to 8 (Figure 5A). Considering the concomitant acidification of water due to increased $CO_2$, future pH levels could affect the P-acquisition mechanisms in cyanobacteria versus heterotrophic bacteria (Figure 5C,D). Since the $CO_2$ increase did not affect the growth of *Halothece* sp. (as they buffered pH), the reduction of APA in cyanobacterial cells could be due to the reduced activity of CCMs, and they would be using the higher $CO_2$ levels to conserve energy [14]. However, low pH may result in fewer free phosphate ions being transported into the cells [80], leading to an increase in APA (and an increase in inorganic P availability), as shown by the increase in APA for *Cobetia* sp. at eCO2 values only at 24 °C (Figure 5C). This suggests the role of temperature in modulating the responses of APA.

Indeed, we found an interesting trend where APA can be inhibited above 24 °C (Figure 5A). In contrast to our in vivo results, in vitro studies using purified alkaline phosphatase show that for cyanobacteria, e.g., *N. flagelliforme*, and heterotrophic bacteria, e.g., *C. amphilecti* KMM 296, the optimum APA is in the range of 40 to 45 °C [77,79,81]. To date, this is the first study reporting the in vivo temperature effect of the APA in marine $N_2$ fixing bacteria (Figure 5B). It is important to consider that the stability of the alkaline phosphatase enzyme, the velocity of organic phosphate breakdown, and enzyme-substrate affinity are controlled by temperature [82].

### 4.3.2. Oxidative Stress

Reactive oxygen species (ROS) are biochemical biomarkers used in marine plants and animals [83,84], and they can be used to evaluate the plasticity of the bacterial community to various environmental factors [85,86]. ROS are reactive molecules whose accumulation, if not removed, can lead to oxidative stress, and thus affect cell viability [86]. Previous studies show that the accumulation of ROS can affect growth and impair $N_2$ fixation activities in the strains tested [35]. Here, we found that cell abundance correlates negatively with the production of ROS (Figure 7B). This implies, as does APA, that the production of ROS can be used as a biomarker to assess the health status of diazotrophic microorganisms exposed to climate change factors.

In agreement with our results, in other studies in phytoplankton cells, e.g., *Chattonella marina* and *Heterosigma akashiwo*, alkaline pH increases ROS production [87,88]. Yet, our results indicate species-specific ROS production responses to alkaline pH, since for *Cobetia* sp. higher pH increased ROS production (Figure 6A). We also observed the same trend for the temperature experiments, i.e., species-specific responses, and cooler or warmer temperatures can promote adaptive responses to survive under suboptimal conditions, which can trigger higher ROS production [88,89] (Figure 6B). Interestingly, our results showed an interactive effect between $CO_2$ and Fe, increasing the ROS levels for *Cobetia* sp. High Fe levels can enhance Fenton and Haber-Weiss reactions, which generate free radicals such as hydroxyl radicals ($\cdot OH$), which are extremely toxic for cells [90]. On the other hand, $CO_2$ can downregulate the catalase expression, which is responsible for the detoxification of the $H_2O_2$, causing a 6-fold decrease in ROS removal [91]. Therefore, both $CO_2$ and Fe can increase ROS production in such a way that $N_2$ fixing heterotrophic bacteria cannot compensate with their antioxidant defenses.

### 5. Conclusions

We demonstrated the vulnerability of $N_2$ fixers found in association with the Mediterranean seagrass *Posidonia oceanica* to ocean acidification and warming due to elevated $CO_2$. However, the effects may depend on whether the bacteria are phototrophic or heterotrophic and on nutrient status, suggesting that the community structure of the $N_2$ fixing community

may also change. Our results reveal the susceptibility of the heterotrophic bacteria. Here, we suggest the utilization of the P-acquisitions mechanisms, i.e., APA, and biochemical parameters, i.e., ROS, as molecular biomarkers to evaluate the responses of diazotrophic cells to climate change factors. These results should be taken with caution, because we tested only some $N_2$ fixers found in association with *P. oceanica*. Ocean warming can increase $N_2$ fixation rates in *P. oceanica* [32], but further studies need to be conducted in natural communities to determine how these $N_2$ fixers respond to climate change and how this may affect plant health.

**Supplementary Materials:** The following supporting information can be downloaded at: https://www.mdpi.com/article/10.3390/d15030316/s1, Table S1: List of all treatments, strains and variables studied in this report. * Only tested for *Halothece* sp. and *Cobetia* sp.; Figure S1: Experimental set-up of the $CO_2$ experiments, considering two different levels of $CO_2$, a$CO_2$: 410 ppm and elevated, e$CO_2$: 1000 ppm, and included a control with no $CO_2$ influx. The pH level was monitored for a$CO_2$ and e$CO_2$; Figure S2: pH monitoring under a continuous influx of atmospheric $CO_2$ 410 ppm (atmospheric $CO_2$, a$CO_2$) and elevated $CO_2$ 1000 ppm (elevated $CO_2$, e$CO_2$) for *Halothece* sp. and *Cobetia* sp.

**Author Contributions:** V.F.-J. conducted all experiments with the help of E.H.Z., E.P.-P. and N.S.R.A. in the various parameters measured in the study. V.F.-J. and N.S.R.A. led the writing of the MS, and N.S.R.A. is the supervisor of the laboratory. All authors have read and agreed to the published version of the manuscript.

**Funding:** This work was supported by funding to NSRA through the Ministerio de Economía, Industria y Competitividad–Agencia Estatal de Investigación and the European Regional Development Funds project (CTM2016-75457-P).

**Institutional Review Board Statement:** Not applicable.

**Data Availability Statement:** Not applicable.

**Acknowledgments:** We acknowledge the support and help of the Scientific Technical Service (Guillem Ramis Munar and Maria Trinidad Garcia Barceló) of the University of the Balearic Islands (UIB) for cytometry and gas-chromatography analyses, respectively. We acknowledge Aida Frank Comas and Xabier López-Alforja for the help in the $CO_2$ set-up. We also thank Pere Ferriol Buñola, Alba Coma Ninot, and Group de Microbiologia de la UIB for the help in the acquisition of the diazotrophic cultures.

**Conflicts of Interest:** The authors declare that the research was conducted in the absence of any commercial or financial relationships that could be construed as a potential conflict of interest.

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
