# Peer review of "Cell Plasticity of Marine Mediterranean Diazotrophs to Climate Change Factors and Nutrient Regimes"

_diversity, doi:10.3390/d15030316_

Round 1
Reviewer 1 Report
The studies the individual responses of marine diazotrophic microorganisms are very interesting in times of global climate change. The studies were well designed, carried out, written and referenced. The figures are adequate but some require a few minor technical adjustments. The manuscript could be accepted for publishing after minor revision.
Detail comments
References e.g. No. 32, 36 - unnecessarily capital letters used in article titles
References No. 53, 90 - authors' full names are given instead of their initials
Fig. 1B, Fig. 3ABC, Fig. 4AB - minor correction of Y-axis captions is needed
Author Response
We appreciate all the recommendations and comments on the manuscript. We corrected the minor changes in the references and the Y -axis captions in Fig. 1B, Fig. 3ABC, Fig. 4AB.
Reviewer 2 Report
Title: Phenotypic plasticity of N2 fixing bacteria to pH changes, temperature, and elevated CO2 in combination with different nutrient regimes- Effects of climate change factors in diazotrophs.
The authors evaluated the phenotypic plasticity of four culturable diazotrophs (i.e., cyanobacteria and heterotrophic bacteria), found in association with the endemic Mediterranean seagrass Posidonia oceanica in combination with different nutrient regime. Overall, the topic is interesting, timely and a relevant research. I found some of the description of the paper to be detailed, and the description and explanation of some of the important points are well presented. Minor review comments must be addressed first to accept this manuscript for publication. Also, the novel point in this study should be emphasized. Why is this study important and how is this research different from other previously reported ones.
Minor Comments:
1) Please strongly justify the novelty of this study. How different is it from previous studies done using the same method. Also, may I request the author(s) to provide a brief summary of previous studies done similar to this study (in the introduction and discussion part). Voucher samples should have been deposited in a recognized Herbarium for documentation purposes.
2) The author should explain their basis of choosing the treatments (ranges) for pH, temperature, and CO2 in the experimental set up.
3) The author should also document the morphological changes in the microbial cells (especially for cyanobacteria) as part of phenotypic plasticity to improve the results and discussion part of the manuscript. I suggest that the authors provide SEM images of the cells to document and describe these changes.
3) Please double check and correct all graphs and figures present in the manuscript. Some of the axis title is not properly labeled.
Author Response
We appreciate all the recommendations and comments on the manuscript, and please see below our responses.
1) Please strongly justify the novelty of this study. How different is it from previous studies done using the same method. Also, may I request the author(s) to provide a brief summary of previous studies done similar to this study (in the introduction and discussion part). Voucher samples should have been deposited in a recognized Herbarium for documentation purposes.
We try to justify the novelty of this study in the abstract (lines 44-47) and introduction (lines 135-137). Also, we highlight the use of APA and ROS in the introduction (lines 114-121), and discussion (lines 561-563, 596-601), presenting a brief summary. We are aware that the summary is from our previous studies, but there very is little information about APA and ROS in diazotrophs.
Regarding the voucher samples. We are not completely sure what the reviewer means but are not working with the plant itself, just with cultures that can be found in association with Posidonia oceanica. It is true that all the bacteria tested can be found in bacterial culture collections, and anyone can buy them and test them.
2) The author should explain their basis of choosing the treatments (ranges) for pH, temperature, and CO2 in the experimental set up.
Temperature explanation was added in the lines (188-189). pH ranges were pointed in the lines (183-184), and for the CO2 values as well pointed out in the lines as atmospheric levels (410 ppm), and predicted (1000 pm) levels (lines 53-56, 201-203).
3) The author should also document the morphological changes in the microbial cells (especially for cyanobacteria) as part of phenotypic plasticity to improve the results and discussion part of the manuscript. I suggest that the authors provide SEM images of the cells to document and describe these changes.
This is a very good point that we did not address. As the reviewer says, it is well-reported that changes, especially in temperature, can change the shape and size of bacteria. Unfortunately, we did not have available a SEM to track these changes, and now it is not possible to carry out that analysis. We tried to compare differences by flow cytometry using the SSC and FSC data, but no real pattern was detected. Therefore, in the discussion, we added that temperature is a factor modulating that change but that this was not addressed (Lines 483-486).
4) Please double check and correct all graphs and figures present in the manuscript. Some of the axis title is not properly labeled.
Thank you for letting us know. We think some axes are not properly labeled because the format changed when we submitted the word. In the original word, they are properly labeled. We will try to fix this in the next submission. Please, see the pdf version to avoid changes in the format.
Reviewer 3 Report
The manuscript is well written and the experimental conditions are well described.So this manuscript contains interesting and publishable data.
Just a small note that will need to be mentioned in the discussion or conclusion
sections. What could be the positive or negative effects of these findings on the
host organism (Posidonia oceanica)?
Author Response
We appreciate all the recommendations and comments on the manuscript. Please, see below our responses.
What could be the positive or negative effects of these findings on the host organism (Posidonia oceanica)?
The effects on the plant only based on culture based-approaches are really difficult to extrapolate to what is really happening in the plant. And this is something to address in further studies. Actually, we made some studies showing as these factors can change the diazotrophic activity, in this case increasing the rates but potentially changing the diazotrophic structure, as referenced in the lines 633-635.
Reviewer 4 Report
Diversity
Title: Phenotypic plasticity of N2 fixing bacteria to pH changes, temperature, and elevated CO2 in combination with different nutrient regimes – Effects of climate change factors in diazotrophs
Author: Victor Fernandez-Juarez
General Comments
The study examines how marine nitrogen-fixing microorganisms might respond to nutrients, temperature, pH, and CO2 in sea water. This is not a new idea. But the studies, here. are interesting.
I have a few concerns. On is the difference between the present study and your previous studies. The author does cite those studies, but several sentences suggest the data presented here are also in previous papers. Perhaps it is due to the writing style, but please clarify
Another concern is Statistical Analyses. These need to be described in more detail; see below.
Specific Comments
1) Title is a bit long. Consider ‘Effects of climate change factors on marine diazotrophs from Mediterranean seagrass.’
2) The Abstract could be better. For example, give specific results. Rather than saying ‘we show how sensitive, etc.’ state the findings that show the degree of sensitivity. Also, the last sentence should be more specific: state the nutrients, levels, and lifestyle.
3) The Introduction reads well. One suggestion is adding a hypothesis or expected result to the last paragraph. For example, did you expect the four organisms to show similar responses to each other?
4) One suggestion on writing. Typically, phrases introduced by ‘e.g.,’ and ‘i.e.,’ go at the end of sentence. For example, ‘e.g.,’ introduces a list, and having a list in a middle of a sentence is a bit awkward. There are many places in the manuscript.
5) Also, do not put the i.e.-phrase in parentheses. It is part of the sentence, not to be buried in parentheses.
6) I suppose it is okay to adjust acidic pH with HCl. However, you are adding a lot of chloride. Did you consider impact of chloride on the cultures?
7) It is important in the Methods when you describe experiments, especially, those with different pH and temperature to indicate the length of the experimental period. it is important to know if cells were growing or at steady state when the measurements were made.
8) In the Statistical Analysis section, start by describing the experimental design. Give the main effects in the model and how you handled interactions. The full model has pH, nutrient, temperature, CO2, and species. With all the interactions your power of test is limited.
9) In general, the Results seem logical to me.
10) The Discussion is interesting.
Technical Comments
1) Line 13: delete, ‘i.e., bacteria capable to reduce N2 to ammonium.’
2) Line 55: add a comma before ‘in general.’
3) Line 62: consider deleting the comma before ‘by reducing.’
4) Line 141: delete ‘for the pH experiment.’
5) Line 239: delete ‘clear.’ Regressions are ‘significant’, or not.
6) Figure 1A: is the relationship for nutrient limited cells non-linear, or is this just a curved line fit?
7) Line 331: change ‘on the contrary’ to ‘conversely.’
8) Figure 4 A & B; are these non-linear or just a function of the line fit graphing program?
9) Line 603: is this true. Iron speciation occurs but mostly at pH values much more acidic than in seawater.
Author Response
We appreciate all the recommendations and comments on the manuscript. Please, see below our responses.
The author does cite those studies, but several sentences suggest the data presented here are also in previous papers. Perhaps it is due to the writing style, but please clarify.
We re-wrote this section, lines (107-114).
1) Title is a bit long. Consider ‘Effects of climate change factors on marine diazotrophs from Mediterranean seagrass.’
We tried to reduce the tittle and we changed to: “Cell plasticity of marine diazotrophs to climate change factors and nutrient regimes”
We are aware that our work is not novel in the field, but the strong points are that we compared the physiological changes between cyanobacteria and heterotrophic bacteria, that is barely known in the field, and then we investigated biomarkers, through changes in the cell plasticity, to study changes in the diazotrophic population. Therefore, we think this is important to highlight in the title.
2) The Abstract could be better. For example, give specific results. Rather than saying ‘we show how sensitive, etc.’ state the findings that show the degree of sensitivity. Also, the last sentence should be more specific: state the nutrients, levels, and lifestyle.
We re-wrote the abstract trying to state the main findings.
3) The Introduction reads well. One suggestion is adding a hypothesis or expected result to the last paragraph. For example, did you expect the four organisms to show similar responses to each other?
We added the expected results/hypothesis in the lines (119-122).
4) One suggestion on writing. Typically, phrases introduced by ‘e.g.,’ and ‘i.e.,’ go at the end of sentence. For example, ‘e.g.,’ introduces a list, and having a list in a middle of a sentence is a bit awkward. There are many places in the manuscript. 5) Also, do not put the i.e.-phrase in parentheses. It is part of the sentence, not to be buried in parentheses.
Thank you for the suggestion. We tried to correct this through the text.
6) I suppose it is okay to adjust acidic pH with HCl. However, you are adding a lot of chloride. Did you consider impact of chloride on the cultures?
The reviewer is right, we did not consider the impact of Cl on the cultures. However, Cl is a micronutrient for normal cell function, and at least should not have a deleterious effect on the cells. There are only few papers that address the effect of Cl in cultures, but according to RoeMler et al. 2003, Cl is NOT required for growing.
7) It is important in the Methods when you describe experiments, especially, those with different pH and temperature to indicate the length of the experimental period. it is important to know if cells were growing or at steady state when the measurements were made.
In the section with different pH and temperatures, the conditions were not specified, because it was already mentioned in the section "Experimental culture conditions". We clarified for ALL experiments, cells were kept in the original conditions of 120 rpm, 12 h dark: 12 h light, and under low-intensity fluorescent light (30 µE m-2 s-1) for 72 hours, also shown in the Supplementary Table S1. We added the cell phase as well, clarifying that the analyses were carried out in the mid exponential phase (line 176).
8) In the Statistical Analysis section, start by describing the experimental design. Give the main effects in the model and how you handled interactions. The full model has pH, nutrient, temperature, CO2, and species. With all the interactions your power of test is limited.
We tried to clarify the statistical analysis section in lines (273-279). We are aware that we have a complex metadata with pH, nutrient, temperature, CO2, and species, and our statistical analyses are very limited. Here, we wanted to highlight the effect of global climate change factors with different nutrient regimes without entering in stronger statistical analyses, e.g., modeling these interactions, since we have very few strains and representatives of cyanobacteria and heterotrophic bacteria to do that.
Technical Comments
Figure 1A: is the relationship for nutrient limited cells non-linear, or is this just a curved line fit?
It is non-linear regression. This was specified in the lines line 316.
8) Figure 4 A & B; are these non-linear or just a function of the line fit graphing program?
It is just a function of the line fit graphing program. This was pointed out in the figure caption.
9) Line 603: is this true. Iron speciation occurs but mostly at pH values much more acidic than in seawater.
We removed this part of speciation to avoid confusions.